# IL-22 inhibits ferroptosis and attenuates ischemia-reperfusion-induced acute kidney injury: Association with activation of the P62-Keap1-Nrf2 signaling pathway

Lin Zhang[1], Feng Luo[2]*, Yalin Chai[1], Lijie Sun[1], Xuan Wang[1], Le Yin[1], Congjuan Luo[1]*

**1** Department of Nephrology, The Affiliated Hospital of Qingdao University, Qingdao, Shandong Province, China, **2** Department of Cardiology, The Affiliated Hospital of Qingdao University, Qingdao, Shandong Province, China

* luofeng19821124@126.com (FL); luocongjuan2018@163.com (CL)

## Abstract

Acute kidney injury (AKI) remains a major clinical challenge due to its high morbidity and mortality, with ischemia-reperfusion injury (IRI) as one of its primary causes. Severe IRI-associated AKI (IRI-AKI) can progress to irreversible renal failure, yet no effective therapies are currently available. Ferroptosis, an iron-dependent regulated cell death, has recently been implicated in the pathogenesis of IRI-AKI. Moreover, IL-22 may alleviate AKI by modulating the ferroptosis process through regulation of the P62-Keap1-Nrf2 signaling axis. In this study, we examined the protective role of the immune cytokine interleukin-22 (IL-22) in IRI-AKI and its mechanistic association with ferroptosis. Using a murine IRI model and an HK-2 cell hypoxia/reoxygenation system, we systematically assessed the impact of IL-22 treatment. IL-22 administration significantly enhanced renal function, reduced histological injury, and limited both reactive oxygen species accumulation and ferroptotic cell death. Further mechanistic studies demonstrated that IL-22 suppresses ferroptosis in vitro through an Nrf2-dependent mechanism and is associated with activation of the P62-Keap1-Nrf2 signaling pathway. These findings offer experimental evidence supporting IL-22 as a potential therapy for IRI-AKI and highlight ferroptosis modulation as a promising therapeutic strategy.

## 1. Introduction

AKI is a serious clinical syndrome characterized by a sudden decline in kidney function over a short period (hours to weeks), typically manifested by elevated serum creatinine, decreased urine output, and metabolic disturbances [1]. It affects approximately 10–15% of hospitalized patients worldwide, with the incidence rising to more than 50% in intensive care units [2,3]. While AKI has diverse etiologies, IRI is recognized as one of the most common and severe pathogenic mechanisms, typically occurring in clinical contexts such as kidney transplantation, cardiopulmonary

**Data availability statement:** All relevant data are within the paper and its Supporting information files.

**Funding:** The author(s) received no specific funding for this work.

**Competing interests:** The authors have declared that no competing interests exist.

resuscitation, major hemorrhage, and septic shock [4]. IRI consists of an ischemic phase caused by impaired blood supply followed by reperfusion, which induces oxidative stress, mitochondrial dysfunction, and aggravated tubular epithelial cell damage, thereby increasing the likelihood of progression from AKI to chronic kidney disease [5,6]. Therefore, a comprehensive understanding of the pathogenesis of IRI-AKI, particularly its key regulatory nodes, is essential for identifying new therapeutic targets.

Ferroptosis, an iron-dependent form of regulated cell death distinct from apoptosis and necrosis, has emerged as a crucial contributor to tissue injury across multiple organs [7]. It is driven by lipid peroxidation and has been shown to increase markedly during IRI-AKI, with its extent closely linked to the severity of renal damage [8,9]. Experimental evidence indicates that targeting iron metabolism, lipid peroxidation, or antioxidant defense systems can effectively suppress ferroptosis and mitigate kidney injury [10–12]. Among these regulatory pathways, nuclear factor erythroid 2-related factor 2 (Nrf2) plays a central role in maintaining redox balance and ferroptosis resistance. Through activation of antioxidant response elements, Nrf2 upregulates protective proteins such as GPX4 and SLC7A11, which are essential for iron homeostasis and cell survival [13,14]. Several studies have demonstrated that enhancing Nrf2 signaling alleviates ferroptosis-driven renal damage in IRI-AKI [15,16]. Upstream, the adaptor protein P62 (SQSTM1) promotes Nrf2 nuclear translocation by competitively binding to Keap1, thereby activating downstream antioxidant transcriptional programs. This canonical P62-Keap1-Nrf2 axis is now considered a major mechanism regulating ferroptosis and protecting against ischemic tissue injury [17–19].

IL-22 is a member of the IL-10 cytokine family that is predominantly produced by Th17 cells, γδ T cells, and type 3 innate lymphoid cells. It mediates its biological actions via binding to a heterodimeric receptor complex composed of IL-22R1 and IL-10R2 on epithelial cells [20]. Accumulating studies have highlighted the potent tissue-protective capacity of IL-22 in various pathological contexts. Experimental models of hepatic fibrosis have demonstrated that IL-22 enhances the expression of P62, which in turn activates the Keap1-Nrf2 signaling axis, contributing to its anti-oxidative and anti-fibrotic actions [21]. Within renal tissue, IL-22 facilitates tubular epithelial cell regeneration and proliferation, implying its potential to regulate ferroptosis via the P62-Keap1-Nrf2 pathway, thereby mitigating acute kidney injury [22,23]. Nevertheless, the precise molecular functions and mechanistic details of IL-22 in IRI-AKI have yet to be fully elucidated.

In this study, we employed a hypoxia/reoxygenation model in human HK-2 cells and a mouse renal IRI model to systematically evaluate the protective effects of IL-22 in IRI-AKI. We further investigated whether the anti-ferroptotic effects of IL-22 are associated with the P62-Keap1-Nrf2 pathway, thereby providing mechanistic insight and experimental support for targeting ferroptosis in IRI-AKI.

## 2. Materials and methods

### 2.1. Experimental animals and animal models

Male C57BL/6 mice (aged 8–12 weeks) were acquired from Pengyue Laboratory Animal Co., Ltd. (Jinan, China). Housed in a barrier-controlled environment (22 ± 1°C,

55±5% humidity, 12-hour photoperiod), the animals were randomly allocated into three cohorts: (1) sham-operated controls, (2) ischemia-reperfusion (IR) injury group, and (3) IR+IL-22 treatment group. Thirty minutes preoperatively, mice in the IL-22 intervention cohort received 1 mg/kg recombinant IL-22 protein via intraperitoneal administration, while counterparts in the remaining groups were injected with equal volumes of saline. Under anesthesia with pentobarbital sodium (50 mg/kg, intraperitoneally), the left renal pedicle was clamped for 40 minutes, and the right kidney was removed to enhance compensatory stress, thus establishing a unilateral renal ischemia-reperfusion model. Sham-operated mice underwent the same surgical procedure without vascular occlusion. During the surgery, the mice were placed on a thermostatically controlled heating pad to prevent hypothermia; postoperative thermal support was maintained for 24 hours to promote reperfusion recovery, and the animals were monitored twice daily. After 24 hours of reperfusion, the mice were re-anesthetized with pentobarbital sodium (50 mg/kg, intraperitoneally) until complete loss of reflexes, followed by euthanasia via cervical dislocation. Serum and kidney tissue samples were then immediately collected for functional and histopathological analyses. All experimental procedures were approved by the Animal Ethics Committee of Qingdao University (Approval ID: QYFY WZLL 30510).

## 2.2. Cell culture and cell model

The human renal proximal tubular epithelial cell line HK-2 (Procell Life Science & Technology, Wuhan, China) was cultured in a complete growth medium containing 10% fetal bovine serum (Gibco) and 1% penicillin-streptomycin (Solarbio). Cells were maintained at 37°C in a humidified incubator with 5% $CO_2$. Cells were stratified into four experimental arms: (1) control group, (2) hypoxia-reoxygenation (HR) injury, (3) HR+IL-22 overexpression, and (4) HR+IL-22+ML385 cotreatment. To simulate HR injury in vitro, confluent cultures (80% density) were transferred to glucose- and serum-deprived DMEM/F12 medium and exposed to a low-oxygen environment (1% $O_2$) for 24 hours, followed by 12-hour reoxygenation under normoxic conditions. For pharmacological interventions, IL-22-group cells were pretreated with 100 ng/mL recombinant IL-22 protein for 3 hours prior to HR induction. The ML385 cotreatment cohort underwent sequential incubation with 20 μmol/L ML385 (1 hour) and IL-22 (3 hours) before hypoxia exposure.

## 2.3. qRT-PCR

Total RNA was isolated from renal tissues or cultured cells using TRIzon lysis reagent (Invitrogen). RNA purification involved chloroform/isopropanol phase separation followed by centrifugation, with integrity verified spectrophotometrically. For cDNA synthesis, 2 μg RNA underwent reverse transcription with the HiScript cDNA synthesis kit (Vazyme), under optimized thermal conditions. Amplification reactions were conducted on a Roche LightCycler 96 platform using SYBR Green I master mix and target-specific primers. Transcript quantification employed the $2^{-\Delta\Delta Ct}$ calculation method, normalized against β-actin as an endogenous control. The primer sequences utilized are detailed in S1 Table.

## 2.4. Western blot

Cellular lysates were prepared from renal specimens and cultured cells using ice-cold lysis buffer supplemented with protease/phosphatase inhibitors, followed by 30-minute agitation at 4°C. Cellular debris was pelleted by centrifugation (12000 g, 15 min), with the resultant supernatant retained for protein quantification via colorimetric assay. Protein aliquots were combined with 5×Laemmli buffer and subjected to thermal denaturation (95°C, 5 min) prior to electrophoretic separation. Electrophoresis commenced at 80 V through the stacking phase, transitioning to 120 V for resolution in the separating matrix. Subsequent protein transfer to methanol-treated PVDF membranes was conducted. Membranes were blocked with 5% skim milk prior to sequential incubation with primary antibodies (4°C, overnight) and HRP-linked secondary antibodies (25°C, 60 min), interspersed by three TBST washes (10 min each). Chemiluminescent signals were captured using enhanced substrate detection, with band densitometry normalized to GAPDH expression. The primary antibodies used were: GPX4 (1:1000, Proteintech), ACSL4 (1:1000, Proteintech), SLC7A11 (1:1000, Abcam), P62 (1:1000, Proteintech),

Keap1 (1:1000, Abcam), Nrf2 (1:1000, Proteintech), and GAPDH (1:50000, Proteintech). The secondary antibodies were goat anti-rabbit or goat anti-mouse IgG-HRP (1:10000, ZB-2301, China).

## 2.5. Flow cytometry for ROS detection

Processed cells from each group were collected and resuspended in 100 µL of PBS. They were then incubated in the dark at room temperature for 30 minutes with the DCFHDA probe (50101ES01, Yeasen, Shanghai, China). Fluorescent signals were subsequently collected using the NovoCyte 2060R flow cytometer (Agilent, CA, USA), and lipid reactive oxygen species levels were analyzed using FlowJo software (version 10.8.1).

## 2.6. ELISA

Glutathione (GSH) and malondialdehyde (MDA) levels were quantified using commercially available ELISA kits (Elabscience Biotechnology). Tissue homogenates or cellular extracts were diluted in phosphate-buffered saline. Following primary incubation, 50 µL of horseradish peroxidase conjugates were introduced, with subsequent repetition of thermal equilibration and washing protocols. Chromogenic development involved sequential administration of tetramethylbenzidine and hydrogen peroxide substrates. Absorbance values were recorded on a BioTek Synergy H1 microplate reader using dual-wavelength spectrophotometric measurement.

## 2.7. Renal function assessment

Serum creatinine (SCr) and blood urea nitrogen (BUN) concentrations were determined using a commercial detection kit (Nanjing Jiancheng Bioengineering Institute, Nanjing, China). According to the supplier's protocol, serum samples obtained 24 hours after surgery were analyzed colorimetrically with an automated biochemical analyzer.

## 2.8. Histology

Renal tissues underwent fixation in 4% paraformaldehyde (4°C, 24 hr) followed by PBS rinses (0.1 M, pH 7.4) and archival preservation in ethanol. Specimens were processed through graded ethanol dehydration, paraffin embedding, and microtome sectioning (4–8 µm). Conventional H&E staining and glycogen-targeting PAS methodology were employed to evaluate cortical architecture and tubular integrity. The assessment of morphological changes in renal tubules included four key parameters: 1) loss of the brush border structure; 2) dilation or rupture of the tubule lumen; 3) deposition of protein casts; and 4) sloughing of epithelial cells. A semiquantitative scoring system was established as follows: Grade 0 (no lesions), Grade 1 (involvement≤25%), Grade 2 (26–50%), Grade 3 (51–75%), and Grade 4 (>75%). All tissue sections were evaluated blindly by two or three pathologists to ensure objectivity.

## 2.9. Transmission electron microscopy

Specimens were initially immersed in glutaraldehyde-based fixative (4°C, 24 hr) and sequentially rinsed with phosphate buffer. Membrane contrast enhancement was achieved through osmium tetroxide post-fixation (1%, 2 hr, 25°C). Processed specimens underwent graded ethanol series dehydration (50–100%) prior to resin infiltration. Embedding protocols included transitional immersion in epoxy-acetone mixtures (1:1 v/v, 12 hr) followed by pure epoxy resin saturation. Polymerization occurred at 60°C over 48 hr. Ultrathin sections (40–60 nm) were generated using a diamond-tipped ultramicrotome. Double-contrast staining employed uranyl acetate (15 min) and lead citrate (15 min) before ultrastructural visualization using a Hitachi HT7700 TEM.

## 2.10. Statistical analysis

Triplicate experimental replicates generated datasets expressed as arithmetic mean ± SD. Comparative analyses were executed through GraphPad Prism (v10.1.2). Parametric evaluations incorporated Student's t-test and one-way ANOVA.

Post hoc multiplicity adjustment via Bonferroni correction ensured stringent control over type I error inflation during inter-group comparisons.

## 3. Results

### 3.1. IL-22 improves renal function and alleviates tissue injury in an IR injury model

To investigate whether IL-22 confers renal protection during IRI-AKI, a murine model of renal IR injury was generated. Subsequently, the influence of IL-22 on renal function and histopathological alterations was comprehensively evaluated in vivo. Serum biochemical analyses revealed that compared to the Sham group, mice in the IR group exhibited significantly elevated levels of SCr and BUN, indicating severe impairment of kidney function. However, following treatment with recombinant IL-22 protein, both markers showed a significant decline, suggesting a notable protective effect of IL-22 on renal function (Fig 1A). Further histological assessment using H&E staining revealed structural changes in the renal tissue. The IR group showed visible signs of injury, including vacuolar degeneration of tubular epithelial cells, tubular casts, and interstitial hemorrhage, with pathology scores significantly higher than those of the Sham group. In contrast, IL-22 treatment markedly alleviated the degree of pathological changes, leading to significantly lower scores, indicating that IL-22 facilitates the mitigation of tissue damage induced by IR injury (Fig 1B). Additionally, IR stimulation significantly upregulated the mRNA expression levels of kidney injury markers KIM-1 and NGAL, while treatment with IL-22 resulted in a significant downregulation of these markers, further supporting its role in alleviating tubular injury (Fig 1C). Collectively, these results demonstrate that IL-22 can effectively mitigate renal damage caused by ischemia-reperfusion through reducing injury to tubular epithelial cells and improving kidney function.

### 3.2. IL-22 attenuates IR-induced renal ferroptosis in mice

Growing evidence has established ferroptosis as a pivotal mechanism underlying IRI-AKI. To determine whether IL-22 confers renal protection by modulating this process, we examined molecular markers associated with ferroptosis. qRT-PCR analyses revealed that renal tissues from IR mice exhibited significantly elevated ACSL4 mRNA, a promoter of ferroptosis, alongside decreased expression of the inhibitory factors GPX4, SLC7A11, and Nrf2. Treatment with IL-22 effectively reversed this expression profile, suggesting suppression of ferroptosis (Fig 2A). Western blot analyses corroborated these findings at the protein level. IR kidneys showed diminished GPX4, SLC7A11, and Nrf2, with increased ACSL4, whereas IL-22 administration normalized these alterations (Fig 2B).

Since lipid peroxidation is a hallmark of ferroptosis, we next examined markers of oxidative stress. Using MDA and GSH assay kits, we found that MDA levels were significantly higher and GSH levels markedly lower in the IR group compared with the sham group, indicating enhanced oxidative stress. Importantly, IL-22 treatment significantly reversed these changes, suggesting its protective role against oxidative stress (Fig 2C). Because mitochondrial damage is another defining feature of ferroptosis, we performed transmission electron microscopy. Compared with Sham mice, IR kidneys displayed shrunken mitochondria with reduced cristae and disrupted membranes. In contrast, IL-22 treatment largely preserved mitochondrial morphology, maintaining cristae integrity (Fig 2D). Collectively, these results show that IL-22 suppresses IR-induced ferroptosis through multiple mechanisms, including restoration of ferroptosis-inhibiting factors, reduction of lipid peroxidation, and preservation of mitochondrial structure.

### 3.3. IL-22 treatment correlates with activation of the P62-Keap1-Nrf2 pathway in mouse kidney tissue

Previous studies demonstrate that Nrf2 protects against oxidative stress, and the P62-Keap1-Nrf2 axis is recognized as a critical pathway in mitigating IR injury. Under physiological conditions, Keap1 binds to Nrf2 and promotes its degradation, thereby maintaining low Nrf2 levels. Notably, P62 competitively binds to Keap1, disrupting the Keap1-Nrf2 complex and releasing Nrf2. After release, Nrf2 translocates into the nucleus, where it binds to AREs and initiates transcription of antioxidant and cytoprotective genes, including key ferroptosis inhibitors such as GPX4 and SLC7A11. This mechanism removes lipid peroxides, preserves redox balance, and prevents ferroptosis.

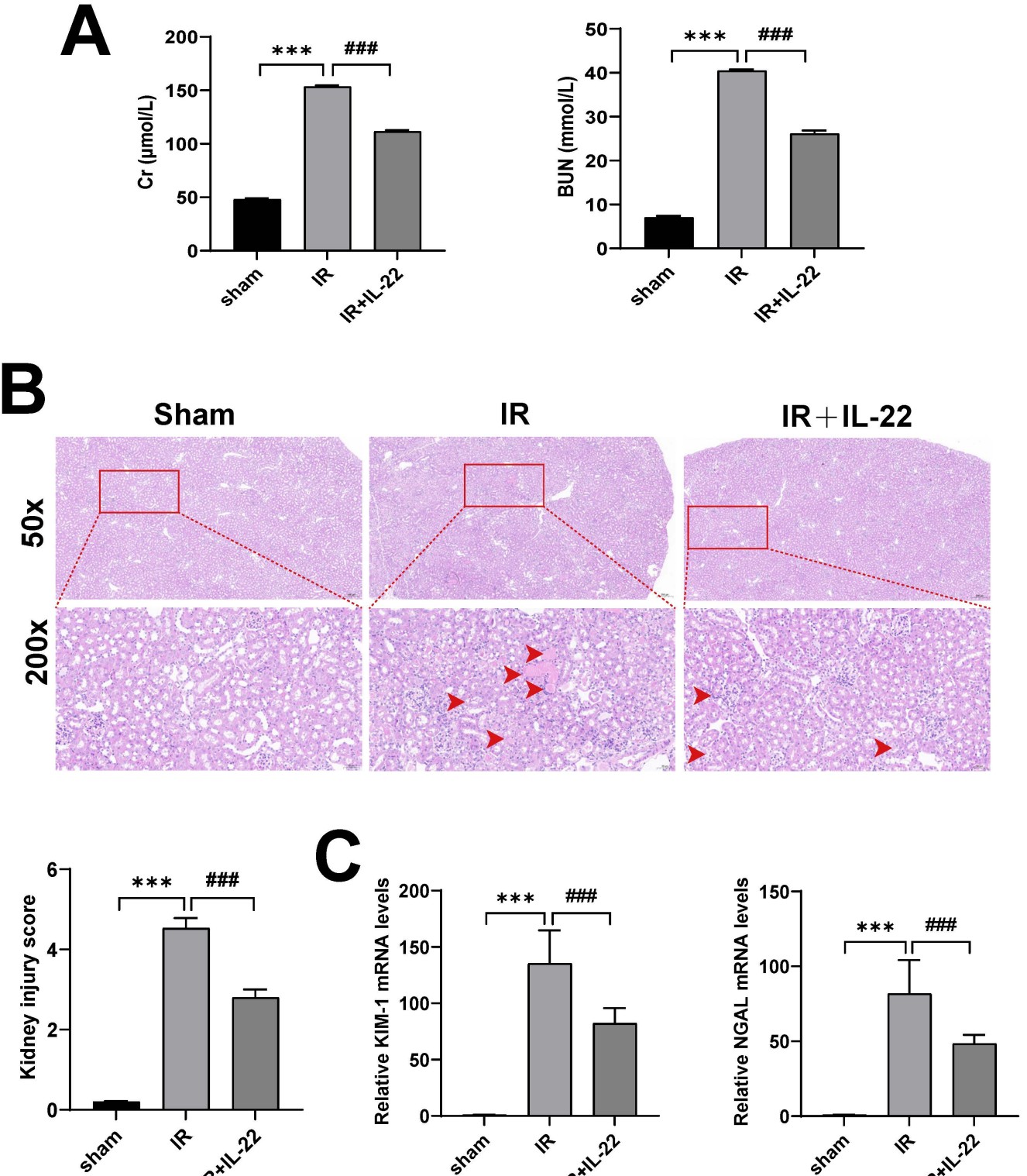

**Fig 1. IL-22 alleviates ischemia-reperfusion-induced kidney injury in mice.** (A) serum creatinine (SCr) and blood urea nitrogen (BUN) levels; (B) renal histology by H&E staining and injury scores; (C) renal mRNA expression levels of KIM-1 and NGAL. Data are presented as mean ± SD, n = 3. ***$p < 0.001$ vs. the Sham group; ###$p < 0.001$ vs. the IR group.

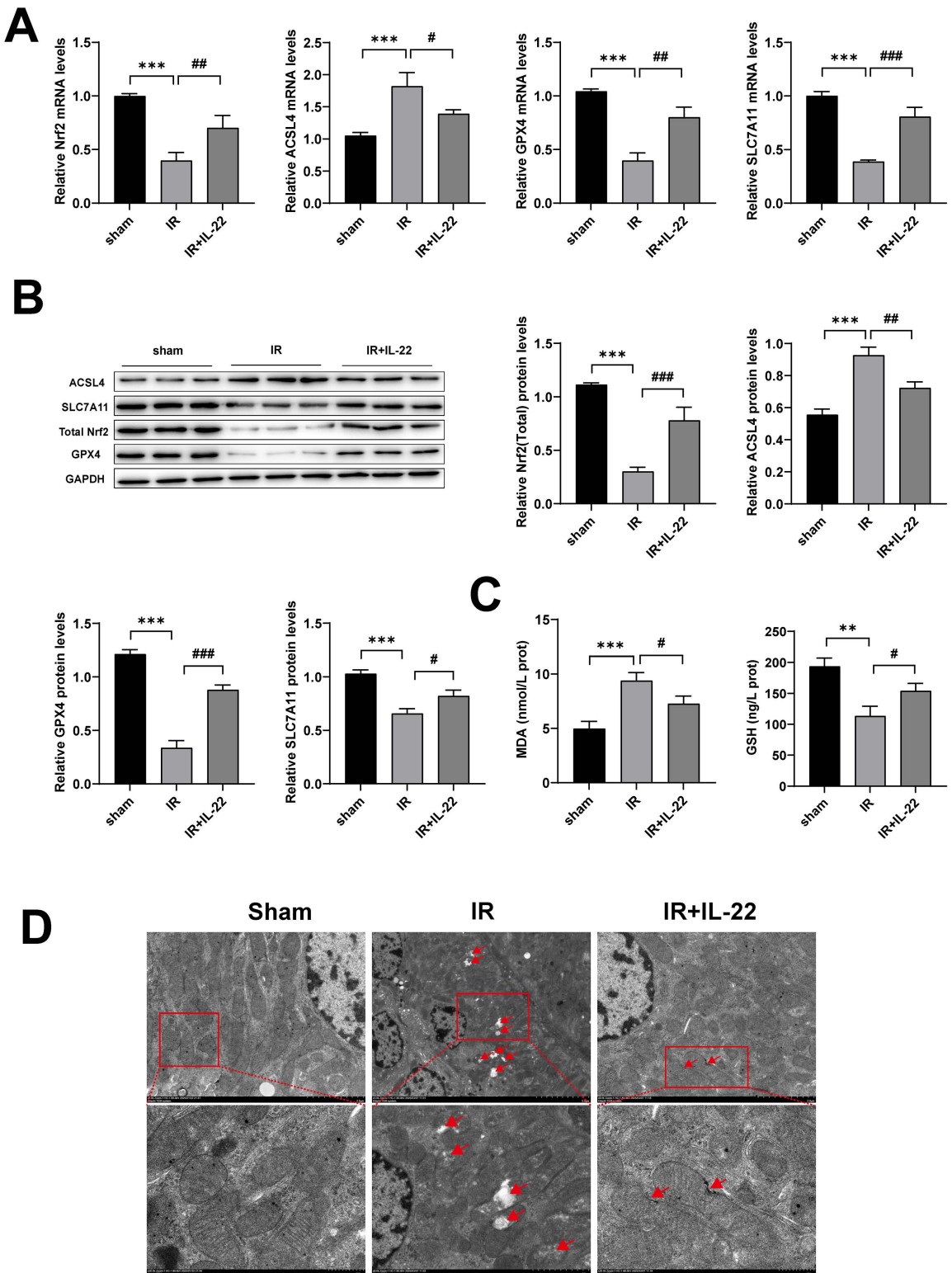

**Fig 2. IL-22 suppresses renal ferroptosis.** (A) mRNA levels of total Nrf2, ACSL4, GPX4, and SLC7A11 in kidney tissues; (B) protein expression of total Nrf2, ACSL4, GPX4, and SLC7A11 in different groups; (C) renal levels of MDA and GSH measured by ELISA; (D) mitochondrial morphology of renal tubular epithelial cells (Scale bar, 5 and 2 μm). Data are presented as mean ± SD, n = 3. **p < 0.01, ***p < 0.001 vs. the Sham group; #P < 0.05, ##P < 0.01, ###p < 0.001 vs. the IR group.

To determine whether IL-22 exerts its effects through this signaling pathway, we examined the expression levels of key components of the P62-Keap1-Nrf2 axis. qPCR and Western blot analyses revealed that, compared with the IR group, kidney tissues from the IR+IL-22 group exhibited significantly increased levels of P62 and total Nrf2, along with a pronounced reduction in Keap1 expression (Fig 3A, B). Furthermore, Nrf2 nuclear translocation was markedly enhanced (Fig 3C). These findings demonstrate that IL-22 treatment is significantly associated with activation of the P62-Keap1-Nrf2 pathway in mouse kidney tissue. However, whether IL-22 directly mediates ferroptosis inhibition and renal protection in vivo via this pathway remains to be confirmed through further functional studies.

### 3.4. IL-22 suppresses HR-induced ferroptosis in HK-2 cells

In our in vitro experiments, we established a HR model using HK-2 cells to examine the regulatory effects of IL-22 on ferroptosis at the cellular level. Consistent results from qRT-PCR and Western blot analyses showed that HR treatment caused abnormal expression of ferroptosis-related molecules: ACSL4 was significantly elevated, while levels of GPX4, SLC7A11, and Nrf2 were decreased; however, IL-22 treatment effectively reversed these changes (Fig 4A, B).

Extensive evidence indicates that excessive iron-dependent lipid peroxidation is a major contributor to HK-2 cell injury. Flow cytometry showed that HR stimulation significantly increased ROS levels in HK-2 cells, reflecting heightened oxidative stress. In contrast, IL-22 treatment markedly reduced ROS accumulation (Fig 4C). ELISA further confirmed that IL-22 increased GSH levels and decreased MDA levels, thereby alleviating lipid peroxidation (Fig 4D). Furthermore, we examined the mitochondrial structure using transmission electron microscopy, which revealed severe mitochondrial damage in HR-treated cells, characterized by typical features of ferroptosis (Fig 4E). Conversely, IL-22 treatment partially protected mitochondrial structure, preserving functional integrity. These in vitro results further confirm that IL-22 effectively suppresses ferroptosis, improves cellular oxidative status, and maintains mitochondrial homeostasis in the cellular model.

### 3.5. IL-22 alleviates HR-induced ferroptosis via Nrf2 signaling

To determine whether the cytoprotective effect of IL-22 against ferroptosis depends on Nrf2, we employed the selective Nrf2 inhibitor ML385 in the subsequent experiments. ML385 is a well-established small-molecule inhibitor of Nrf2 that binds to its transcriptional activation domain, thereby preventing Nrf2 from interacting with ARE sequences and inhibiting Nrf2-dependent transcriptional activity [24]. Notably, Nrf2 transcriptionally regulates P62. Thus, ML385 not only attenuates Nrf2-mediated regulation of downstream antioxidant and anti-ferroptotic genes but may also disrupt P62 expression, indirectly affecting the stability of the Keap1-Nrf2 signaling axis [25]. qPCR and Western blot analyses revealed that IL-22 treatment decreased Keap1 expression, increased P62 and total Nrf2 levels, and promoted Nrf2 translocation into the nucleus, leading to enhanced nuclear accumulation. However, ML385 partially inhibited these effects. Specifically, ML385 treatment significantly reduced nuclear Nrf2 levels, increased cytoplasmic Keap1 expression, and substantially decreased the expression of P62 and total Nrf2 in the cytoplasm (Fig 5A–C).

Further analysis of Nrf2 downstream targets involved in ferroptosis showed that IL-22 markedly upregulated the expression of ferroptosis-inhibitory genes GPX4 and SLC7A11, while downregulating the ferroptosis-promoting genes ACSL4. Upon ML385 intervention, the IR+IL-22+ML385 group exhibited a significant increase in ACSL4 expression, accompanied by a substantial decrease in GPX4 and SLC7A11 levels (Fig 5D, E). These results suggest that in the HR-induced HK-2 cell model, IL-22's anti-ferroptotic effect depends on Nrf2 signaling and is closely associated with regulation of the P62-Keap1-Nrf2 axis.

In summary, IL-22 significantly attenuates IR- or HR-induced ferroptosis in both animal and cellular models. In vivo, this protective effect correlates with activation of the P62-Keap1-Nrf2 signaling axis. In vitro mechanistic studies further confirm that Nrf2 plays a critical role in mediating IL-22's anti-ferroptotic effects. Collectively, these findings provide experimental evidence supporting IL-22 as a potential therapeutic strategy.

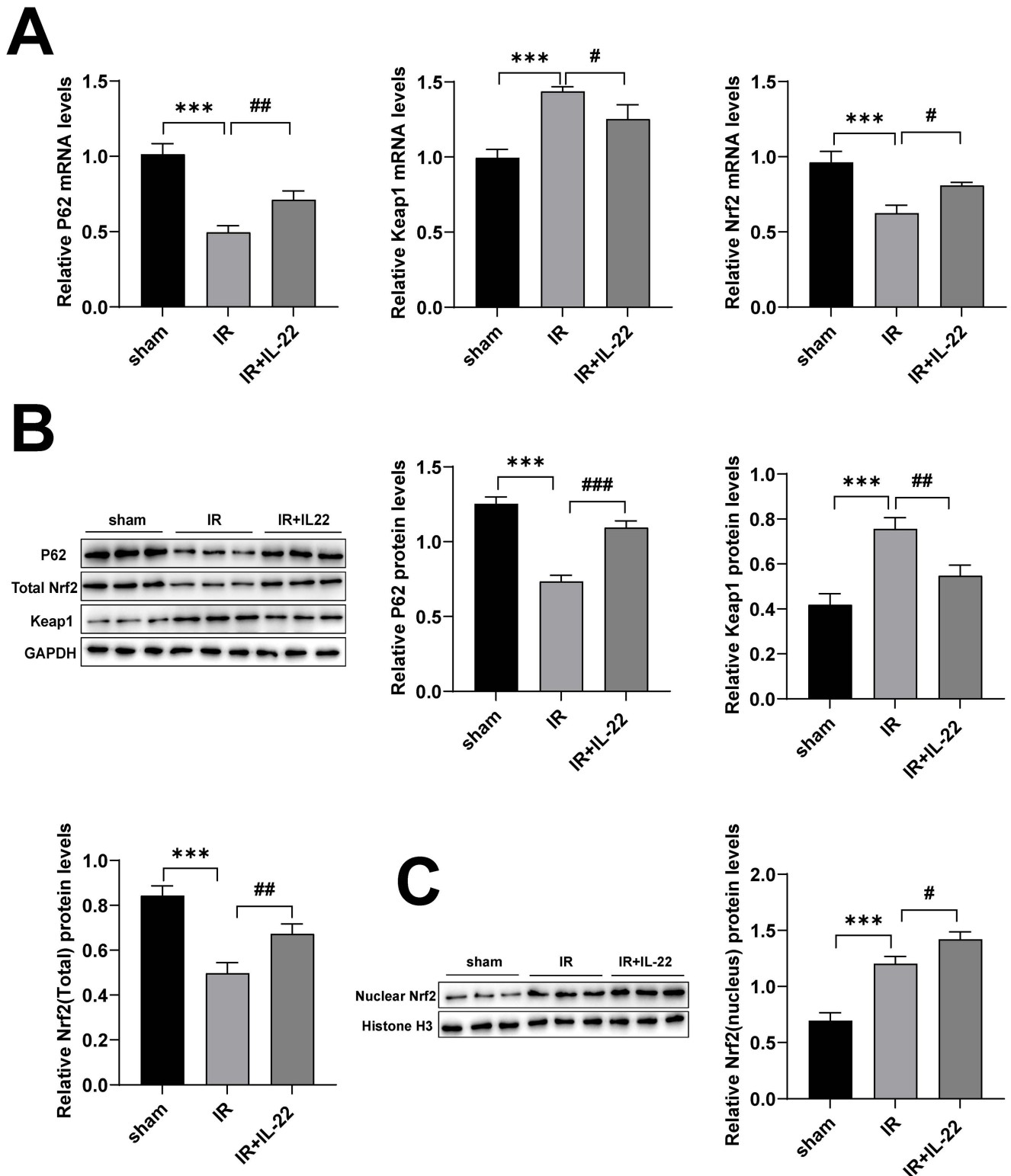

**Fig 3. IL-22 regulates the P62-Keap1-Nrf2 pathway.** (A) mRNA levels of P62, Keap1, and total Nrf2 in renal tissues from different groups; (B) protein levels of P62, Keap1, and total Nrf2 in kidney tissues; (C) nuclear Nrf2 protein expression in mouse kidneys. Data are presented as mean ± SD, n = 3. ***p < 0.001 vs. the Sham group; #P < 0.05, ##P < 0.01, ###p < 0.001 vs. the IR group.

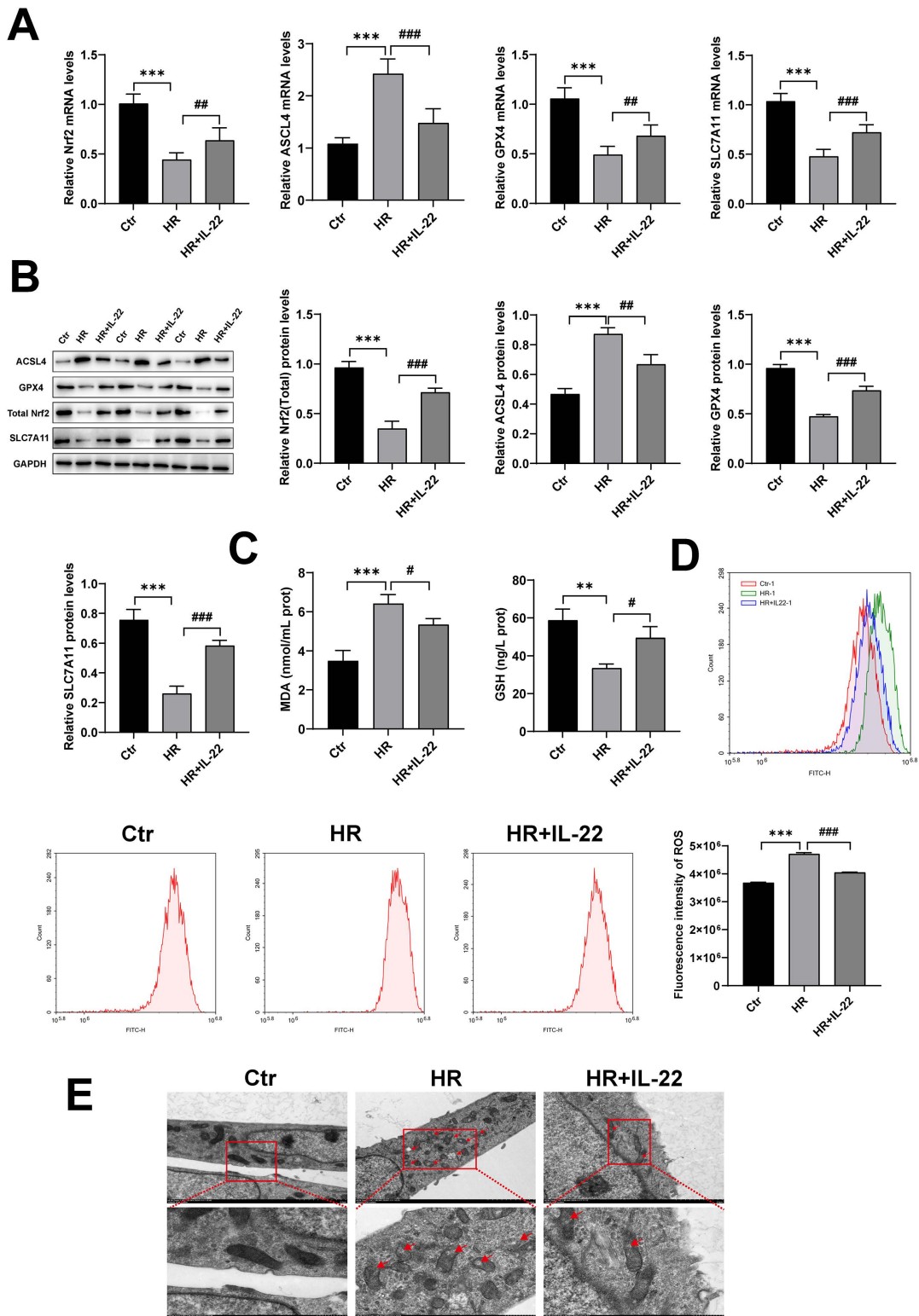

**Fig 4. IL-22 attenuates ferroptosis in HK-2 cells.** (A) mRNA levels of total Nrf2, ACSL4, GPX4, and SLC7A11 in different cell groups; (B) protein expression of total Nrf2, ACSL4, GPX4, and SLC7A11 in cell lysates; (C) levels of MDA and GSH in cell lysates measured by ELISA; (D) ROS levels in different groups detected by flow cytometry; (E) mitochondrial morphology of cells in each group (Scale bar, 5 and 2 μm). Data are presented as mean±SD, n=3. **p<0.01, ***p<0.001 vs. the Ctr group; #P<0.05, ##P<0.01, ###p<0.001 vs. the HR group.

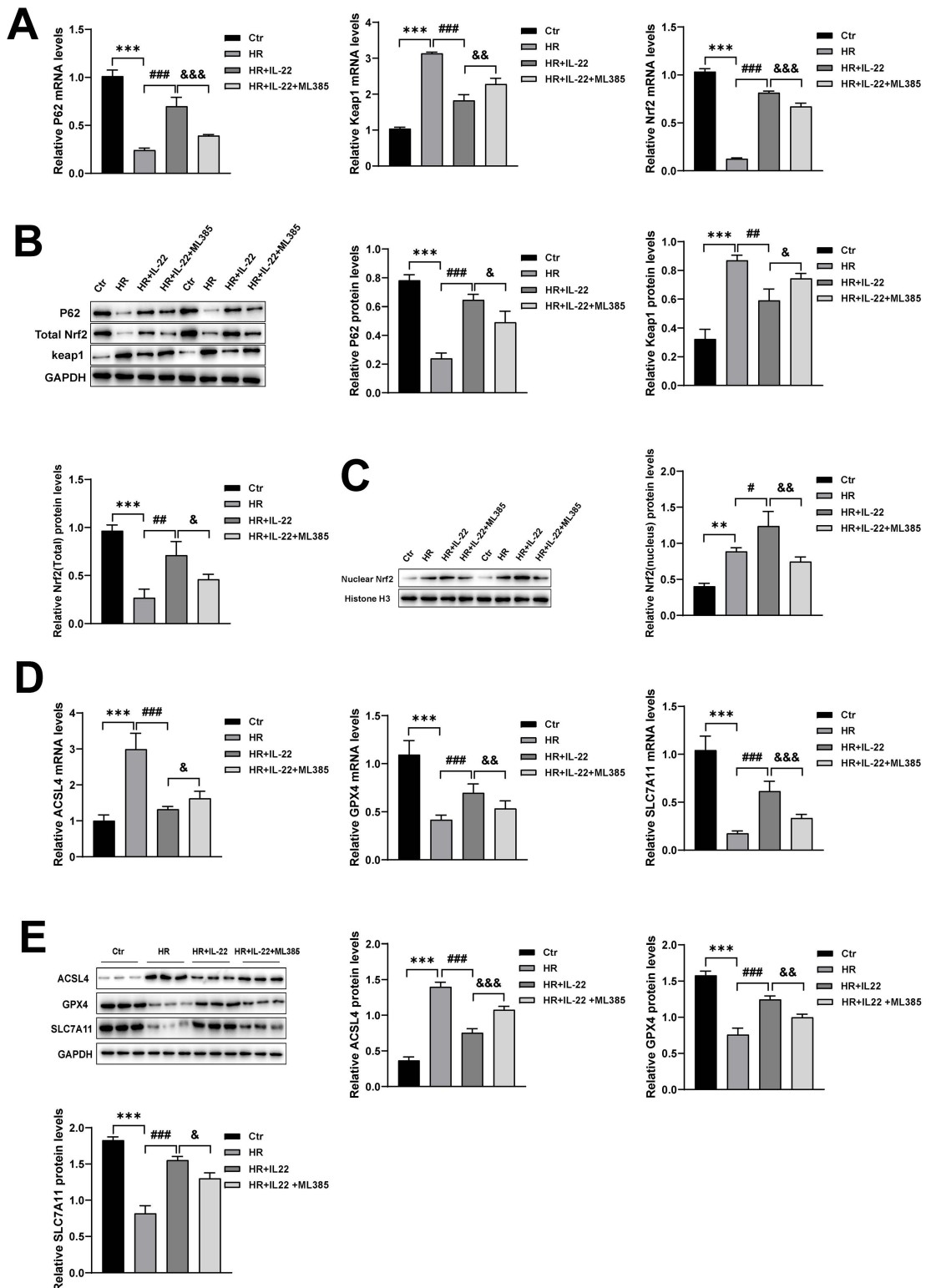

**Fig 5. IL-22 inhibits ferroptosis by modulating the P62-Keap1-Nrf2 pathway.** (A) mRNA levels of P62, Keap1, and total Nrf2 in different cell groups; (B) protein levels of P62, Keap1, and total Nrf2 in cell lysates; (C) nuclear Nrf2 protein expression in different groups; (D) mRNA levels of ACSL4, GPX4, and SLC7A11 in different cell groups; (E) protein expression of ACSL4, GPX4, and SLC7A11 in cell lysates. Data are presented as mean ± SD, n = 3. **p < 0.01, ***p < 0.001 vs. the Ctr group; #P < 0.05, ##P < 0.01, ###p < 0.001 vs. the HR group; &P < 0.05, &&P < 0.01, &&&p < 0.001 vs. the HR + IL-22 group.

## 4. Discussion

IRI is a major pathological driver of AKI, especially in the context of renal transplantation, and is strongly associated with poor clinical outcomes [26]. Current therapeutic strategies are limited to supportive measures such as continuous renal replacement therapy (CRRT), as no specific pharmacological agents are available for IRI-induced AKI. This gap highlights the urgent need for novel therapeutic approaches [27]. In this study, we used a mouse IRI model and a HR model in HK-2 cells to demonstrate that IL-22 markedly inhibits ferroptosis and mitigates IRI-AKI (Fig 6), highlighting a potential therapeutic target for AKI.

The renoprotective effects of IL-22 in IRI-AKI are correlated with activation of the P62-Keap1-Nrf2 signaling axis and increased nuclear translocation of Nrf2. Following nuclear translocation, Nrf2 binds to antioxidant response elements (AREs), leading to increased expression of antioxidant and anti-ferroptotic factors, accompanied by suppression of ferroptosis-promoting molecules. In contrast, the selective Nrf2 inhibitor ML385 partially inhibits Nrf2-dependent transcriptional activity, thereby attenuating the anti-ferroptotic effects mediated by IL-22. This figure is an original schematic illustration created entirely by the authors and is provided for illustrative purposes only.

IL-22 is a cytokine primarily secreted by immune cells, broadly expressed in various human tissues and organs, including the liver, lungs, kidneys, skin, pancreas, and gastrointestinal tract. Its biological functions significantly depend on the local microenvironment and disease pathology, displaying notable duality [28–30]. Research indicates that persistent or excessive IL-22 signaling is closely associated with various chronic inflammatory diseases and may accelerate the development of certain tumors by promoting epithelial cell proliferation and inhibiting apoptosis [31–33]. However, despite its potential pathogenic effects, IL-22 has been widely recognized for its protective role in acute injury repair and barrier function maintenance. In contexts of acute injury, IL-22 has been shown to reduce oxidative stress, suppress apoptosis, and accelerate epithelial repair, thereby limiting tissue damage in organs such as the liver, intestine, and lung [28,34–36]. However, its role in IRI-AKI and the underlying mechanisms have not been fully clarified.

Recent studies increasingly support the potential value of IL-22 in kidney protection. IL-22 predominantly targets proximal tubular epithelial cells and activates signaling pathways such as STAT3, AKT, and AMPK to reduce mitochondrial

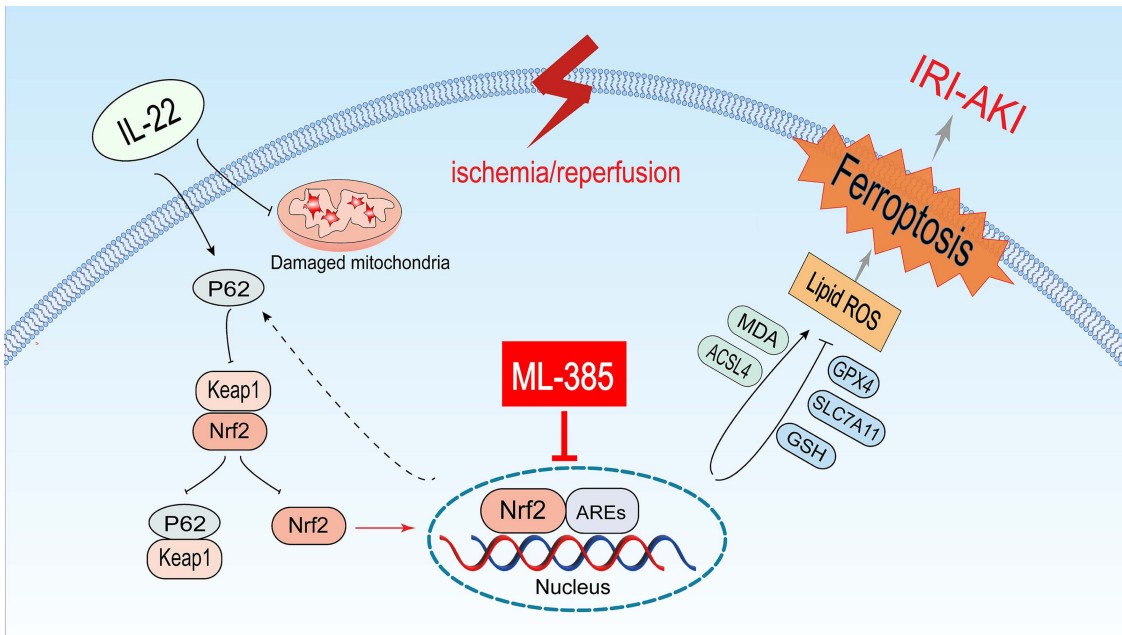

**Fig 6. Schematic illustration of the mechanisms by which IL-22 alleviates IRI-AKI.**

damage and oxidative stress, promoting cell survival and regeneration [22,23,37,38]. In various animal models, including those of cisplatin-induced kidney injury, diabetic nephropathy, and acetaminophen-related AKI, IL-22 has exhibited roles in promoting metabolic reprogramming, maintaining mitochondrial homeostasis, and inhibiting cell death [22,39]. In addition, Taghavi and colleagues recently demonstrated that IL-22 maintains endothelial glycocalyx stability, providing further evidence for its therapeutic potential in trauma-associated AKI [40]. Together, these findings suggest that IL-22 holds promise as a new strategy for treating IRI-AKI.

In our investigation, administration of IL-22 led to a notable decline in serum BUN, SCr, KIM-1, and NGAL concentrations in mice, along with a substantial improvement in renal tissue morphology and reduced histological damage. These results suggest that IL-22 confers significant renoprotective effects by preserving renal function and mitigating kidney injury caused by ischemia-reperfusion insult. To gain deeper insight into the molecular basis of this protection, we examined whether IL-22 mediates its effects by modulating ferroptosis—an iron-dependent and lipid peroxidation-driven mode of regulated cell death that is critically involved in the development of IRI-AKI [8,41]. Earlier investigations have shown that suppressing ferroptosis through pharmacological agents or genetic modulation effectively attenuates renal injury linked to ischemia-reperfusion events [17,42,43]. Consistent with these reports, our experiments in both cell and animal models demonstrated that IL-22 markedly inhibited ferroptosis and substantially decreased lipid peroxidation related to this pathway.

To explore the potential mechanism underlying IL-22-mediated regulation of ferroptosis, this study further focused on its possible effects on the P62-Keap1-Nrf2 signaling pathway. Under basal conditions, Keap1 targets Nrf2 for degradation. Upon redox stimulation, P62 competitively binds Keap1, releasing Nrf2 and promoting its nuclear translocation. This process induces the expression of ARE-dependent genes, thereby regulating anti-ferroptotic molecules including GPX4 and SLC7A11 [18,44]. In our study, IL-22 treatment increased P62 expression, reduced Keap1 levels, and enhanced nuclear accumulation of Nrf2. Correspondingly, the expression of anti-ferroptotic molecules GPX4 and SLC7A11 was upregulated, while pro-ferroptotic factors such as ACSL4 were downregulated. Consistent with previous reports showing that Entacapone and Urolithin A alleviate IRI-AKI via activation of the P62-Keap1-Nrf2 pathway [15,45], our results suggest that IL-22 may exerts cytoprotective effects under ischemia-reperfusion conditions by modulating ferroptosis-related processes, potentially via the P62-Keap1-Nrf2 signaling axis.

Nevertheless, this study has several limitations. In the HK-2 cell model, the Nrf2-specific inhibitor ML385 reversed IL-22's inhibition of ferroptosis, indicating that Nrf2 activation mediates the anti-ferroptotic effects of IL-22. However, as P62 knockdown or silencing was not performed, the observed changes in P62 and Keap1 expression levels only suggest a potential association between Nrf2 activation and the upstream P62-Keap1 pathway, and the direct regulatory relationship between these components remains to be confirmed. Moreover, mechanistic validation in this study was primarily based on in vitro experiments, without in vivo genetic or pharmacological interventions targeting P62. Therefore, the present findings mainly indicate an association between the renoprotective effects of IL-22 in vivo and activation of the P62-Keap1-Nrf2 axis, rather than a direct causal relationship. Future studies employing in vivo P62 knockout models or targeted intervention strategies will be required to determine whether the protective effects of IL-22 on the kidney depend on this signaling axis. Notably, IL-22 has been reported to induce cell death through activation of DNA damage responses under certain conditions, suggesting that its renal effects may vary depending on the disease stage and microenvironment. Therefore, a more systematic exploration of IL-22 activity across different pathological contexts is warranted [28,46]. In addition, the lack of recognized specific biomarkers for ferroptosis limits a comprehensive evaluation of the therapeutic effects of IL-22. Developing sensitive and specific ferroptosis markers will be crucial for advancing the clinical translation of this therapeutic strategy [47].

In summary, this study demonstrates that IL-22 confers significant renoprotective effects in IRI-AKI. IL-22 treatment not only reduced ferroptosis-related lipid peroxidation and enhanced antioxidant defenses but also activated the P62-Keap1-Nrf2 signaling pathway. Collectively, these findings provide experimental evidence supporting IL-22 as a potential therapeutic agent for IRI-AKI and suggest new directions and candidate strategies for targeting ferroptosis in AKI therapy.

## Supporting information

**S1 Table. Primers Used in qRT-PCR.**
(DOCX)

**S1 Raw Images. Western blot raw images for Figs 2–5.**
(PDF)

**S1 Dataset. This dataset contains the raw data used for statistical analysis in this study.**
(ZIP)

## Author contributions

**Conceptualization:** Lin Zhang, Feng Luo, Xuan Wang.

**Data curation:** Lin Zhang.

**Formal analysis:** Lin Zhang, Yalin Chai.

**Methodology:** Congjuan Luo.

**Resources:** Congjuan Luo.

**Software:** Feng Luo, Lijie Sun, Le Yin.

**Supervision:** Feng Luo, Congjuan Luo.

**Validation:** Yalin Chai.

**Visualization:** Feng Luo, Yalin Chai, Lijie Sun, Xuan Wang, Le Yin.

**Writing – original draft:** Lin Zhang, Congjuan Luo.

**Writing – review & editing:** Lin Zhang, Congjuan Luo.

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
