## [Decision Letter · Decision Letter 0]

10 Dec 2025

Thank you for submitting your manuscript to PLOS ONE. After careful consideration, we feel that it has merit but does not fully meet PLOS ONE’s publication criteria as it currently stands. Therefore, we invite you to submit a revised version of the manuscript that addresses the points raised during the review process.

Although both reviewers considered the manuscript worthy of publication, one reviewer raised the concern that the authors should temper their conclusions. Specifically, the presented data show only a correlation between IL-22 elevation and activation of the p62-Keap1-Nrf2 pathway, without demonstrating causality.

We look forward to receiving your revised manuscript.

Kind regards,

Hiroyasu Nakano, M.D., Ph.D.

Academic Editor

PLOS One

Journal Requirements:

3. We note that Figures 1 and 6 in your submission contain copyrighted images. All PLOS content is published under the Creative Commons Attribution License (CC BY 4.0), which means that the manuscript, images, and Supporting Information files will be freely available online, and any third party is permitted to access, download, copy, distribute, and use these materials in any way, even commercially, with proper attribution. For more information, see our copyright guidelines: http://journals.plos.org/plosone/s/licenses-and-copyright.

1. You may seek permission from the original copyright holder of Figures 1 and 6 to publish the content specifically under the CC BY 4.0 license.

4. Please include captions for your Supporting Information files (Original data.rar at the end of your manuscript, and update any in-text citations to match accordingly. Please see our Supporting Information guidelines for more information: http://journals.plos.org/plosone/s/supporting-information.

Reviewers' comments:

Reviewer's Responses to Questions

**Comments to the Author**

1. Is the manuscript technically sound, and do the data support the conclusions?

Reviewer #1: Yes

Reviewer #2: Partly

2. Has the statistical analysis been performed appropriately and rigorously?

Reviewer #1: Yes

Reviewer #2: Yes

3. Have the authors made all data underlying the findings in their manuscript fully available?

Reviewer #1: Yes

Reviewer #2: Yes

4. Is the manuscript presented in an intelligible fashion and written in standard English?

Reviewer #1: Yes

Reviewer #2: Yes

Reviewer #1: Zhang et al. systematically evaluated the effects of IL-22 treatment using a mouse ischemia-reperfusion injury (IRI) model and a hypoxia/reoxygenation system in HK-2 cells. IL-22 administration markedly improved renal function, reduced histological damage, and suppressed cell death mediated by reactive oxygen species accumulation and ferroptosis. Mechanistic analysis revealed that IL-22 activates the P62-Keap1-Nrf2 signaling axis, enhancing downstream antioxidant defense mechanisms. Notably, Nrf2 inhibition by ML385 markedly attenuated these protective effects, emphasizing the essentiality of this pathway.

In summary, these findings demonstrate that IL-22 mitigates IRI-induced acute kidney injury (AKI) by suppressing ferroptosis via the P62-Keap1-Nrf2 cascade. This provides preclinical evidence that IL-22 may represent a potential therapeutic strategy and identifies a promising target for ferroptosis regulation.

Although accumulating evidence has already implicated that IL-22 may alleviate AKI by modulating the ferroptosis through the regulation of the P62-Keap1-Nrf2 signaling axis, as pointed out by the authors, the authors have clearly demonstrated the precise molecular functions and mechanistic details of IL-22 in IRI-AKI.

Overall, the experiments were performed with great rigor, leading to a clear conclusion. This manuscript is also well written. Therefore, I feel that it is worthy of publication even without any revision.

Reviewer #2: The authors investigate the protective role of IL-22 in ischemia-reperfusion injury (IRI)-induced AKI. Using murine IRI and HK-2 cell models, the study shows that IL-22 improves renal function and suppresses ferroptosis markers. The authors conclude these effects are mediated via the p62-Keap1-Nrf2 pathway. The study addresses a significant topic, and the experimental design is generally sound. However, the conclusion that the p62-Keap1-Nrf2 pathway mediates the protection is not fully supported by the data, as mechanistic validation was limited. The authors should temper their conclusions to reflect the limitations described below.

Major Comments

1. The authors conclude that IL-22 functions via the p62-Keap1-Nrf2 axis. However, the mechanistic validation relied solely on ML385, which targets Nrf2 directly. While this confirms the necessity of Nrf2, it does not prove the specific causal involvement of the upstream p62 or Keap1 regulation. The observed changes in p62 and Keap1 expression levels demonstrate a correlation, but without genetic manipulation (e.g., p62 knockdown/silencing), the direct link remains unproven. The authors should acknowledge that the specific role of the p62-Keap1 axis upstream of Nrf2 is suggested by expression data but not functionally confirmed.

2. The conclusion that the p62-Keap1-Nrf2 pathway mediates the in vivo protection is not fully supported by the data, as mechanistic validation was limited to in vitro studies. The claim that IL-22 protects against AKI via this pathway in mice is not supported by causal data (e.g., in vivo inhibitor or knockout studies). The authors must temper this conclusion in the manuscript, clarifying that the in vivo data shows correlation rather than causation, and explicitly state this as a limitation in the Discussion.

**Do you want your identity to be public for this peer review?** For information about this choice, including consent withdrawal, please see our Privacy Policy

Reviewer #1: No

Reviewer #2: No

---

## [Author Response · Author response to Decision Letter 1]

25 Dec 2025

Dear Editor,

We sincerely thank you and the reviewers for taking the time to carefully evaluate our manuscript and for providing constructive comments and suggestions.

In this rebuttal letter, we provide a point-by-point response to all comments raised by the editor and reviewers. We first address the editorial and journal requirement comments, followed by responses to Reviewer #1 and Reviewer #2 comments. Each comment is reproduced below, followed by our corresponding response.

Response to the Academic Editor / Journal Requirements

Comment 1 (Editor):

Response:

Thank you for your guidance regarding the PLOS ONE formatting requirements. We have carefully revised the manuscript to fully comply with PLOS ONE’s style guidelines, including those for file naming, manuscript structure, and formatting, in accordance with the provided PLOS ONE style templates for the main body and title/authors/affiliations. We believe that the revised version now meets all formatting and style requirements of PLOS ONE.

Comment 2 (Editor):

PLOS ONE now requires that authors provide the original uncropped and unadjusted images underlying all blot or gel results reported in a submission's figures or Supporting Information files. Please ensure that your figures adhere fully to these guidelines and provide the original underlying images.

Response:

Thank you for your guidance regarding the reporting requirements for blot and gel images. We confirm that all original, uncropped, and unadjusted raw images underlying the blot results have been provided as Supporting Information. Specifically, these data are included in a PDF file entitled “S1_raw_images”. In the revised manuscript, this file is described in the Supporting Information section as: “S1_raw_images. Western blot raw images for Figs 2-5.” (Page 28, line 688)

In addition, we have clearly stated in the Cover Letter that all raw blot images are available and have been uploaded as Supporting Information in the PDF file named “S1_raw_images”. We believe that these revisions fully comply with PLOS PLOS ONE’s blot and gel reporting and figure preparation guidelines.

Comment 3 (Editor):

Figures 1 and 6 in your submission contain copyrighted images. You are required to either obtain permission to publish under CC BY 4.0, or remove/replace the figures.

Response:

Thank you for your comments regarding potential copyright issues related to Figures 1 and 6. We have carefully revised both figures to ensure full compliance with the CC BY 4.0 licensing requirements of PLOS ONE.

For Figure 1, Panel A (the schematic illustration of the animal experimental design) has been completely removed in the revised manuscript. This change does not affect the study results or conclusions. All panel labels, in-text citations, and the figure caption have been updated accordingly. The current Figure 1 includes only Panels B-D, which present original experimental data generated in this study. (Page 12, line 252)

For Figure 6, all content derived from previously published material has been removed. The revised figure now consists exclusively of original schematic elements created by the authors, and the caption has been updated to indicate that the figure is an original schematic illustration provided for illustrative purposes only. (Page 19, line 400)

We confirm that the current versions of Figures 1 and 6 contain only original content and fully comply with the CC BY 4.0 license requirements of PLOS ONE.

Comment 4 (Editor):

Please include captions for your Supporting Information files (Original data.rar) at the end of your manuscript and update any in-text citations to match accordingly.

Response:

Thank you for your helpful suggestion regarding the Supporting Information files. We have revised the Supporting Information accordingly. The previously uploaded compressed file entitled “Original data” has been renamed to “S1 Dataset”; the contents of the file remain unchanged. This file has been uploaded as Supporting Information.

In addition, a corresponding caption has now been included at the end of the revised manuscript in the “Supporting Information” section as follows: “S1 Dataset. This dataset contains the raw data used for statistical analysis in this study.” (Page 28, line 690)

Comment 5 (Editor):

Any changes to the reference list should be mentioned in the rebuttal letter that accompanies your revised manuscript.

Response:

Thank you for your constructive comments. In accordance with the revision suggestions, we have added new content in the manuscript section “3.5. IL-22 alleviates HR-induced ferroptosis via Nrf2 signaling.” To support this new content, we have included the following two references: (Page 17, line 351-358)

24. Singh A, Venkannagari S, Oh KH, Zhang Y-Q, Rohde JM, Liu L, et al. Small Molecule Inhibitor of NRF2 Selectively Intervenes Therapeutic Resistance in KEAP1-Deficient NSCLC Tumors. ACS Chemical Biology. 2016;11(11):3214-25. doi: 10.1021/acschembio.6b00651.

25. Jain A, Lamark T, Sjøttem E, Bowitz Larsen K, Atesoh Awuh J, Øvervatn A, et al. P62/SQSTM1 Is a Target Gene for Transcription Factor NRF2 and Creates a Positive Feedback Loop by Inducing Antioxidant Response Element-driven Gene Transcription. Journal of Biological Chemistry. 2010;285(29):22576-91. doi: 10.1074/jbc.M110.118976.

These references provide support for the experimental conclusions and mechanistic analyses presented in the newly added content. To facilitate identification by the editor and reviewers, we have highlighted these two newly added references in the reference list.

Responses to Reviewers’ Comments

Comment (Reviewer #1):

Overall, the experiments were performed with great rigor, leading to a clear conclusion. This manuscript is also well written. Therefore, I feel that it is worthy of publication even without any revision.

Response:

We sincerely thank you for the positive and insightful comments on our study. We are very pleased that you recognizes our systematic investigation of the molecular mechanisms of IL-22 and its protective effects in both in vitro and in vivo models. We also greatly appreciate the acknowledgment of the rigor of our experimental design and the quality of the manuscript. We sincerely appreciate you for taking the time to carefully evaluate our manuscript and for providing such positive feedback, which greatly encourages us as we further refine and submit the manuscript.

Comment (Reviewer #2):

1. The authors conclude that IL-22 functions via the p62-Keap1-Nrf2 axis. However, the mechanistic validation relied solely on ML385, which targets Nrf2 directly. While this confirms the necessity of Nrf2, it does not prove the specific causal involvement of the upstream p62 or Keap1 regulation. The observed changes in p62 and Keap1 expression levels demonstrate a correlation, but without genetic manipulation (e.g., p62 knockdown/silencing), the direct link remains unproven. The authors should acknowledge that the specific role of the p62-Keap1 axis upstream of Nrf2 is suggested by expression data but not functionally confirmed.

2. The conclusion that the p62-Keap1-Nrf2 pathway mediates the in vivo protection is not fully supported by the data, as mechanistic validation was limited to in vitro studies. The claim that IL-22 protects against AKI via this pathway in mice is not supported by causal data (e.g., in vivo inhibitor or knockout studies). The authors must temper this conclusion in the manuscript, clarifying that the in vivo data shows correlation rather than causation, and explicitly state this as a limitation in the Discussion.

Response:

We sincerely thank you for taking the time and effort to carefully evaluate our study and for providing constructive comments on the mechanistic interpretation. In accordance with these suggestions, we have carefully revised the manuscript and appropriately adjusted the wording of the conclusions to ensure that they are more strictly confined to the scope supported by the available evidence, while clearly articulating the limitations of the study.

Specifically, we now explicitly state that, as P62 knockdown or silencing experiments were not performed, the observed changes in P62 and Keap1 expression levels only suggest a potential association between the P62-Keap1 axis and Nrf2 activation, rather than a direct regulatory relationship. These statements have been revised accordingly in the manuscript and are clearly discussed in the Discussion section. (Page 22, line 462-468)

In addition, we clarify that mechanistic validation in the present study was primarily based on in vitro experiments, without in vivo genetic or pharmacological interventions targeting P62. Therefore, the manuscript now emphasizes that the in vivo findings mainly reflect an association between the renoprotective effects of IL-22 and activation of the P62-Keap1-Nrf2 axis, rather than a direct causal relationship. These limitations are explicitly discussed in the Discussion section, and future studies employing in vivo P62 knockout models or targeted intervention strategies are proposed to further investigate the causal involvement of this signaling axis. (Page 22, line 468-475)

Moreover, all other mechanistic descriptions and related conclusions involving this signaling pathway throughout the manuscript have been revised accordingly to ensure overall consistency and rigor. We believe that these revisions render the mechanistic interpretation more cautious and accurate.

We would like to once again sincerely thank the editor and the reviewers for their time, careful evaluation, and insightful comments. We greatly appreciate the constructive feedback, which has helped us improve the clarity, rigor, and quality of our manuscript. We hope that the revised manuscript now fully addresses all comments and concerns, and we respectfully look forward to its favorable consideration for publication in PLOS ONE.

---

## [Decision Letter · Decision Letter 1]

22 Jan 2026

IL-22 inhibits ferroptosis and attenuates ischemia-reperfusion-induced acute kidney injury: Association with activation of the P62-Keap1-Nrf2 signaling pathway

PONE-D-25-56033R1

Dear Dr. Luo,

We’re pleased to inform you that your manuscript has been judged scientifically suitable for publication and will be formally accepted for publication once it meets all outstanding technical requirements.

Kind regards,

Hiroyasu Nakano, M.D., Ph.D.

Academic Editor

PLOS One

Additional Editor Comments (optional):

Reviewers' comments:

Reviewer's Responses to Questions

**Comments to the Author**

Reviewer #1: (No Response)

Reviewer #2: All comments have been addressed

2. Is the manuscript technically sound, and do the data support the conclusions?

Reviewer #1: Yes

Reviewer #2: Yes

3. Has the statistical analysis been performed appropriately and rigorously?

Reviewer #1: Yes

Reviewer #2: Yes

4. Have the authors made all data underlying the findings in their manuscript fully available?

Reviewer #1: Yes

Reviewer #2: Yes

5. Is the manuscript presented in an intelligible fashion and written in standard English?

Reviewer #1: Yes

Reviewer #2: Yes

Reviewer #1: (No Response)

Reviewer #2: (No Response)

**Do you want your identity to be public for this peer review?** For information about this choice, including consent withdrawal, please see our Privacy Policy

Reviewer #1: No

Reviewer #2: No

---

## [Editor Report · Acceptance letter]

PONE-D-25-56033R1

PLOS One

Dear Dr. Luo,

I'm pleased to inform you that your manuscript has been deemed suitable for publication in PLOS One. Congratulations! Your manuscript is now being handed over to our production team.

Kind regards,

on behalf of

Professor Hiroyasu Nakano

Academic Editor

PLOS One